# In Vitro Anthelmintic Evaluation of *Gliricidia sepium*, *Leucaena leucocephala*, and *Pithecellobium dulce*: Fingerprint Analysis of Extracts by UHPLC-Orbitrap Mass Spectrometry

**DOI:** 10.3390/molecules25133002

**Published:** 2020-06-30

**Authors:** Néstor Romero, Carlos Areche, Jaime Cubides-Cárdenas, Natalia Escobar, Olimpo García-Beltrán, Mario J. Simirgiotis, Ángel Céspedes

**Affiliations:** 1Departamento de Sanidad Animal, Facultad de Medicina Veterinaria y Zootecnia, Universidad del Tolima, Ibagué 730001, Colombia; aecesped@ut.edu.co; 2Departamento de Química, Facultad de Ciencias, Universidad de Chile, Las Palmeras 3425, Nuñoa, Santiago 7800024, Chile; areche@uchile.cl; 3Grupo de Investigación e Innovación en Salud y Bienestar Animal, Laboratorio de Salud Animal, Centro de Investigación Tibaitatá, Agrosavia, Mosquera 250047, Colombia; jcubides@agrosavia.co; 4Facultad de Ciencias Agropecuarias, Universidad de Cundinamarca, Fusagasugá 252212, Colombia; nataliaescobar@ucundinamarca.edu.co; 5Facultad de Ciencias Naturales y Matemáticas, Universidad de Ibagué, Carrera 22 Calle 67, Ibagué 730002, Colombia; jose.garcia@unibague.edu.co; 6Instituto de Farmacia, Universidad Austral de Chile, Valdivia 5090000, Chile; mario.simirgiotis@uach.cl

**Keywords:** *Gliricidia sepium*, *Leucaena leucocephala*, *Pithecellobium dulce*, UHPLC-MS, *anthelminthic*, *Haemonchus contortus*

## Abstract

In the present work, the anthelmintic activity (AA) of ethanolic extracts obtained from *Gliricidia sepium*, *Leucaena leucocephala*, and *Pithecellobium dulce* was evaluated using the third-stage-larval (L_3_) exsheathment inhibition test (LEIT) and egg hatch test (EHT) on *Haemonchus contortus*. Extracts were tested at concentrations of 0.3, 0.6, 1.2, 2.5, 5.0, 10, 20, and 40 mg/mL. The larval exsheathment inhibition (LEI) results showed that *G. sepium* achieved the highest average inhibition of 91.2%, compared with 44.6% for *P. dulce* and 41.0% for *L. leucocephala* at a concentration of 40 mg/mL; the corresponding IC_50_ values were 22.4, 41.7, and 43.3 mg/mL, respectively. The rates of egg hatching inhibition (EHI) at a concentration of 5 mg/mL were 99.5% for *G. sepium*, 64.2% for *P. dulce*, and 54% for *L. leucocephala*; the corresponding IC_50_ values were 1.9 mg/mL for *G. sepium*, 3.9 mg/mL for *P. dulce*, and 4.3 mg/mL for *L. leucocephala*. The species extracts studied here were also analyzed by ultra-high performance liquid chromatography and Orbitrap high resolution mass spectrometry (UHPLC-Q/Orbitrap/MS/MS), resulting in the compounds’ identification associated with AA. Glycosylated flavonoids and methoxyphenols were observed in all three species: fatty acids in *G. sepium* and *P. dulce*; phenylpropanoids, anthraquinone glycosides, amino acids and glycosylated phenolic acids in *G. sepium*; and flavonoids in *L. leucocephala.* Comparatively, *G. sepium* presented a greater diversity of compounds potentially active against the control of gastrointestinal nematodes, which was associated with the results obtained in the applied tests.

## 1. Introduction

Gastrointestinal parasitism is among the factors with the greatest adverse economic impact on the world’s ovine industry [1,2]. Several studies have demonstrated the impact of this pathology on sheep production [3,4,5]. In Colombia, pregnant sheep are the main group reported to be affected by parasitism, with lambs younger than three months being the most susceptible group [6]. Furthermore, parasites of the Trichostrongylidae, Strongylidae, Trichuridae, and Ancylostomatidae families were found at prevalence rates of 92.5%, 48.3%, 12.0%, and 11.0% in sheep in Tolima, Colombia [7]. This is consistent with the results that claim that *H. contortus* is a common parasite in the tropics [8], and has both high biotic potential and large larval loads in pastures.

Attempts at controlling gastrointestinal parasitism by using chemical substances indiscriminately without epidemiological knowledge [9] has given rise to anthelmintic resistance and chemical traces being present in both food and the environment [10]. This anthelmintic resistance has been demonstrated in different parts of the world [11,12,13,14,15,16]. In Colombia, multiple resistance in gastrointestinal nematodes (*H. contortus*, *Trichostrongylus* spp., and *Cooperia* spp.) has been found, particularly in the regions of Cundinamarca and Valle del Cauca [17].

While some researchers found traces of anthelmintics in food products [18,19], others have warned about the risk of antiparasitic agents in the host–parasite relationship in wildlife environments [20], as well as toxic effects on aquatic and terrestrial fauna [21,22]. To help overcome these problems, deworming control options have been explored, including the use of plant extracts as a natural alternative, as they are easily accessible at the locally level and have less of a potential negative effect on the environment [23,24]. In fact, a number of studies have involved in vitro evaluations of the AA of plant extracts and fractions against *H. contortus* [25,26,27,28], but only a few studies have featured the chemical characterization of active extracts. Some studies have shown anthelmintic effects of compounds such as tannins [29,30,31], flavonoids [32], essential oils [33,34,35], phenylpropanoids [36,37], and alkaloids [38,39].

With respect to the three forage species focused on in this study, AA has been demonstrated against gastrointestinal nematodes such as *H. contortus*, *Teladorsagia circumcincta*, *Trichostrongylus colubriformis*, and *Oesophagostomum columbianum*. For example, *G. sepium* was shown to exert an anthelmintic effect in vitro against larvae [40,41,42] and eggs [42,43], while *L. leucocephala* affected larvae [44,45]; moreover, *P. dulce* inhibited larval development and migration [46] and reduced the elimination of eggs in goats in vivo [47]. Given the importance of these plant species as a food source in ruminants due to their nutritional and productive attributes, it is interesting to compare their anthelmintic effects in vitro as well as to identify the main compounds with potential effects on *H. contortus*. The aims of this study were (i) to evaluate the AA of ethanol extracts of forage species collected in dry tropical forest, namely, *G. sepium*, *L. leucocephala*, and *P. dulce*, through LEIT and EHT, and (ii) to identify bioactive compounds with possible anthelmintic activity in all extracts using UHPLC-Q/Orbitrap/MS/MS.

## 2. Results

### 2.1. Identified Compounds with Reported Anthelmintic Activity

A total of 53 compounds were identified in *G. sepium*, 33 in *L. leucocephala*, and 29 in *P. dulce*. The compounds of these three species associated with AA are shown in Table 1. The chromatograms used in the analysis are shown in the Appendix A, as well as the molecular formulas and structures of the compounds linked to AA (Appendix A). In *G. sepium*, a greater number of compounds (*n* = 18) was identified than in *L. leucocephala* (*n* = 14) or *P. dulce* (*n* = 5); likewise, the number of types of compound was higher in *G. sepium* (*n* = 7) than in *L. leucocephala* (*n* = 3) or *P. dulce* (3). Compounds with an anthelmintic effect were identified as glycosylated flavonoids and methoxyphenols in all three species: fatty acids in *G. sepium* and *P. dulce*; phenylpropanoids other than flavonoids, anthraquinonic glycosides, amino acids, and glycosylated phenolic acids in *G. sepium*; and flavonoids in *L. leucocephala*.

### 2.2. Larval Exsheathment Inhibition

Extracts at concentrations of 40, 20, 10, 5, 2.5, 1.25, and 0.6 mg/mL were analyzed in terms of LEI; plant species and concentration levels were analyzed individually, along with how they interacted together. Figure 1A shows a microscopic image of the L_3_
*H. contortus* larvae, both with and without cuticle, detected through this LEI test.

Regarding the effects of plant species, the LEI percentage of *G. sepium* (19.1%) was significantly higher (*p* < 0.05) than that of *P. dulce* or *L. leucocephala* (11.6% and 10.0%, respectively). With respect to the effects of extract concentration, the LEI effect increased with increasing concentration, showing significant differences (*p* < 0.05) when comparing concentrations of 10, 20, and 40 mg/mL. As for the plant–concentration interaction, the percentage of LEI increased with increasing concentration of the extracts, and significant differences (*p* < 0.05) were observed among the inhibitions produced by the 40, 20, and 10 mg/mL concentrations (Table 2). The single highest LEI (91.2%) was observed at a concentration of 40 mg/mL with *G. sepium*, followed by a 45.1% effect at 20 mg/mL of the same plant, with a significant difference (*p* < 0.05) between the two concentrations. In contrast, the effects of *P. dulce* and *L. leucocephala* at a concentration of 40 mg/mL were 44.6% and 41.0%, respectively. The Tween and DMSO controls had the lowest values, showing significant differences (*p* < 0.05) when compared with most of the other treatments.

### 2.3. Egg Hatch Test

In terms of the effects of the different plant species used to compare the anthelmintic effect, there were significant differences (*p* < 0.05) between the EHI average of *G. sepium* extract and those of the *P. dulce* and *L. leucocephala* extracts. The results obtained with *G. sepium* showed the greatest EHI (52.9%), followed by *P. dulce* (38.4%) and *L. leucocephala* (36.2%).

Significant differences (*p* < 0.05) in EHI were identified when comparing between different concentrations. Concentrations of 2.5, 5, and 10 mg/mL produced inhibition percentages of 30.9%, 73.2%, and 100%. Regarding plant–concentration interaction, the highest single EHI (100%) was detected at a concentration of 10 mg/mL, with no significant differences (*p* > 0.05) among the three plant species; at 5 mg/mL, there were significant differences (*p* < 0.05) in the EHI of *G. sepium* when compared with *P. dulce* and *L. leucocephala* (Figure 2). There were also significant differences (*p* < 0.0001) between the DMSO and Tween negative controls, the positive treatments with fenbendazole, and the treatments with extracts.

#### 2.3.1. Morulated Eggs (ME)

Figure 1B shows a microscopic image of *H. contortus* ME found in the EHT. There were significant differences (*p* < 0.05) in average ME concentration among the three plant species; *G. sepium* obtained the single highest level with 47.6%, followed by *P. dulce* with 42.9% and *L. leucocephala* with 38.4%. When comparing the concentrations of the extracts, it was found that at 20, 10, and 5.0 mg/mL there were significant differences (*p* < 0.05) in the ME averages (99.5%, 95.2%, and 45.2%, respectively).

All three plant species showed a marked difference regarding the plant–concentration interaction (*p* < 0.05) at 5.0 mg/mL (75.6%, 43.9%, and 16.3%). No significant differences (*p* >0.05) were observed among the three plant species at concentrations of 20, 10, 2.5, 1.2, 0.6, and 0.3 mg/mL (Figure 3). With respect to the controls, significant differences (*p* < 0.0001) were observed for DMSO, Tween 80, and fenbendazole when compared with the other treatments.

#### 2.3.2. Larvated Eggs (LE)

The effect of the three plant species, the concentration, and the plant–concentration interaction on the percentage of LE of *H. contortus* was analyzed when applying the EHT. Figure 1C shows the *H. contortus* larvated eggs found in the LE test. Regarding the effect of the particular plant species used, *G. sepium* showed the highest LE average percentage at 11.9%, followed by *L. leucocephala* with 7.7% and *P. dulce* with 4.9%, which were significantly different (*p* < 0.05). With regard to the effect of the concentration of plan extract used, the highest LE percentage was produced at 5 mg/mL, followed by 2.5 mg/mL and 10 mg/mL, which were significantly different (*p* < 0.05), giving averages of 30.9%, 14.5%, and 4.7%, respectively.

Regarding the plant–concentration interaction, significant differences (*p* < 0.05) were found between the average LE count of *L. leucocephala* (42.7%) and those of *G. sepium* and *P. dulce* (24% and 25.9%) at 5 mg/mL, while at a concentration of 2.5 mg/mL, there were significant differences (*p* < 0.05) among all three plant species (38.6% with *G. sepium*, 4.6% with *L. leucocephala*, and 0.4% with *P. dulce*). No significant differences were observed (*p* > 0.05) between the average LE obtained at a concentration of 5 mg/mL with *L. leucocephala* and 0.25 mg/mL with *G. sepium*. Similar findings were obtained across all three extracts, where there was a change in the effect on the transition from larvae to LE at lower concentrations and that from LE to ME at higher concentrations (Figure 4).

### 2.4. Inhibitory Concentrations 50 and 99 (IC_50_, IC_99_)

#### 2.4.1. Larval Exsheathment Inhibition (LEI)

In the determination of the IC_50_ and IC_99_ for LEI, it was observed that the extract of *G. sepium* presented the lowest values, indicating its greater effectiveness in inhibiting the process of exsheathment of L_3_ larvae of *H. contortus. L. leucocephala* and *P. dulce* produced similar results at higher concentrations, indicating less effectiveness (Table 3).

#### 2.4.2. Egg Hatching Inhibition (EHI)

The result of the determination of IC_50_ and IC_99_ for EHI showed a similar effect as that described for LEI; lower values were also observed for the extract of *G. sepium*, indicating its greater effectiveness in inhibiting egg hatching of *H. contortus* compared to *L. leucocephala* and *P. dulce* (Table 3).

## 3. Discussion

The present investigation evaluated the AA of plant extracts of the species *G. sepium*, *L. leucocephala*, and *P. dulce* by means of LEIT and IHT in vitro in *H. contortus*. To complement and establish a plausible explanation for the results obtained in the biological tests, analysis of the extracts was performed using UHPLC-Q/Orbitrap/MS/MS. With regard to biological tests *G. sepium* had a greater effect on LEI than *P. dulce* and *L. leucocephala*, with the latter two species showing similar behavior to each other. However, the three extracts produced smaller effects on LEI than those reported in previous studies conducted on *H. contortus* [41]. *G. sepium* also showed the greatest EHI effect, followed by *P. dulce* and *L. leucocephala*. The results obtained for *G. sepium* generally indicated more potent effects than reported in previous studies [42,43,48], unlike the results obtained for *P. dulce* [46] and *L. leucocephala* [49]. The effects on the ME and LE percentages were greatest for *G. sepium*, as demonstrated by the effect of its extract, which inhibited *H. contortus* embryonic development and egg hatching. Additionally, the effects on ME obtained for all three plant species were greater than those observed in previous studies [50,51]. *G. sepium* achieved similar LE results to those reported by Reference [43], and *L. leucocephala* achieved inhibition rates similar to the *Theobroma cacao* [51].

When comparing the IC_50_ values of *L. leucocephala* for LEI [50,51,52], found lower values than those observed in this study. In addition, the IC_50_ of *G. sepium* for EHI in gastrointestinal sheep parasites [53] was lower than that observed here, and lower than that found [42], while a lower IC_50_ was observed [48] when using extracts and fractions of cotyledon and *L. leucocephala* seeds. The differences found in comparison with previous studies could possibly be associated with the effects of environmental conditions on the crops [54,55], plant materials used, plant varieties [56,57], age and origin of the nematodes used [25,58], procedure for obtaining extracts [59], type of solvent [60], and laboratory techniques.

Our results revealed that *G. sepium* exhibited a greater diversity of secondary metabolites with AA (Table 1), followed by *L. leucocephala* and then *P. dulce*. Glycosyl flavonoids and methoxyphenols were present in all three plant species, which may be associated with the results observed in this research. The glycosyl flavonoids identified in *G. sepium* have been reported to show evidence of AA in *H. contortus* [32] and in several gastrointestinal sheep parasites [61]. This activity has also been demonstrated for other plants containing the same metabolites [61,62,63]. Additionally, glycosyl flavonoids identified in *L. leucocephala* have also been found in plant species [64,65,66,67,68,69] in which AA has been demonstrated by tests on *H. contortus* and other gastrointestinal nematodes.

Compounds of the same class as detected in *P. dulce* have also been reported previously in this same species [70], in other plants [71,72,73] for which AA has been demonstrated in vitro [32,74,75], and even specifically in *H. contortus* [75,76]. However, glycosyl flavonoids and their aglycones were shown to have the opposite effects on gastrointestinal nematodes. It was found that plant fractions non-active against *H. contortus* were mainly made up of glycosylated flavonoids, while hydroxycinnamic derivatives were identified in the most active fraction [77]. Dihydrocapsiate, a methoxyphenol, was identified and quantified in four *Capsicum* species; among these was *Capsicum frutescens* [78], for which there is also evidence of biological activity on *Pheretima posthuma* [79] and *Tubifex tubifex* [80]. All of these findings suggest a possible role of this compound in the observed effects of *G. sepium* and *P. dulce*. Similarly, a study on *Rhus natalensis* and the role of tannins and other polyphenols in the elimination of *H. contortus* found that quinic acid derivatives are possibly associated with this process [81], which could also be related to the anthelmintic effect of *L. leucocephala*.

Major fatty acid compounds with AA were identified in *G. sepium* and *P. dulce*, while in *L. leucocepahala* the molecules with such effects were flavonoids and phenylpropanoids other than flavonoids. Meanwhile, anthraquinonic glycosides, amino acids, and glycosylated phenolic acids were found only in the ethanolic extract of *G. sepium*. When the AA results reported in other studies were analyzed with regard to the aforementioned metabolites, it was found that some of them acted individually, while others had greater effects upon binding to other compounds; members of yet another smaller group were found to be part of larger complex molecules. The presence of a second group of compounds with AA in *G. sepium*, as demonstrated in other studies, could be associated with the differences found in the observed in vitro test results.

Regarding fatty acids, such as those identified in *G. sepium*, a few studies have evaluated their anthelmintic effects; some have shown that these fatty acids probably have the ability to act either individually or together. With respect to azelaic acid specifically, identified its possible effect on *Caenorhabditis elegans* [82], which was suggested to be associated with the activity of the steroid enzyme 5 alpha-reductase [83]. Other authors have reported AA against nematodes such as *Meloidogyne incognita* [84], *Oesophagostomum dentatum* [85], *Haemonchus* spp., *Oesophagostomum* spp., and *Trichostrongylus* [86]. Moreover, it can be observed a combined effect of fatty acids and tannins on larvae of *Toxocora canis* [36].

Regarding the flavonoids identified in *L. leucocephala*, AA against *H. contortus* has been observed in plants in which quercetin was identified [61,81,87]. Other studies demonstrated the same activities of quercetin and apigenin [88], and the presence of these flavonoids has also been reported in *Matricaria recutita* [89,90], a plant proven to exert AA [91]. The flavone apigenin has also been shown to exert anthelmintic effects in nematodes such as *Caenorhabditis elegans*. In addition, in a study with 13 flavones, it was shown that apigenin inhibits larval growth [92]. Subsequently, researched the mechanisms associated with *C. elegans* larval growth inhibition, proposing that apigenin acts as a stressor in order to either activate the DAF-16 transcription factor or inhibit DAF-2/insulin signaling [93].

Other compounds identified in *G. sepium* as anthraquinone glycosides, amino acids, and glycosylated phenolic acids have been associated with AA. With respect to anthraquinone compounds, observed the effect against sheep gastrointestinal parasites of two anthraquinone compounds (1.8 dihydroxiantraquinone and 1.2 dihydroxiantraquinone) [94]; more recently, several anthraquinones were successfully tested and it was specifically 1-methyl-2,3,8-trihydroxiantraquinone that was the most active against *Brugia malayi* and *Schistosoma mansoni* [95]. With respect to amino acids, the importance of glutamic acid in the cytotoxic capacity of cyclotides (cyclic proteins) has been reported [96], as well as the biological capacity in *H. contortus* of cycloviolacin-O2, the most powerful anthelmintic cyclotide [97]. With reference to glycosylated phenolic acids, it has been reported that phenylactic acid (R) containing cyclohexadepsipeptides suc as enniatin is highly active in vivo against *H. contortus* in sheep [98,99].

## 4. Materials and Methods

### 4.1. Chemicals

Absolute ethanol, acetic acid, acetonitrile, sterilized distilled water, methanol hypergrade for LC-MS (Merck, Darmstadt, Germany), DMSO (Mallinckrodt Baker, Phillipsburg, Kentucky, USA), Tween 80 (Sigma-Aldrich, St. Louis, MO, USA), 98% fenbendazole (Sigma-Aldrich, St. Louis, MO, USA), and Lugol (Albor Chemicals) were used.

### 4.2. Plant Material

Leaves of *G. sepium, L. leucocephala*, and *P. dulce* were collected at the Universidad del Tolima’s farm “El Recreo,” located in the rural area of Caracolí in the municipality of Guamo (Tolima-Colombia), at 4°00′32.7′′ N (4.009090) and 74°58′51.4′′ W (74.980943). The plants were identified by Prof. Hector Esquivel and the vouchers of the specimens (No. 18329 (*P. dulce*), No. 18330 (*G. sepium*), and No. 18331 (*L. leucocephala*)) are kept in the herbarium of the Universidad del Tolima.

### 4.3. Extraction and Isolation

The leaves of the plants were dried at 25 °C in the dark and ground in a mill (1 kg each). They were then macerated individually in ethanol (three times, 2.0 L, 5 days/extraction), followed by the extracts being filtered by gravity using Whatman filter paper, grade 1, and the solvent being concentrated under reduced pressure at 45 °C, obtaining 9.2%, 4.8%, and 6.1% yields of crude extracts for *G. sepium*, *L. leucocephala*, and *P. dulce*, respectively. UHPLC-Q/Orbitrap/MS/MS analysis of all ethanol extracts was performed in order to identify the predominant compounds.

### 4.4. Instrumentation

A Thermo Scientific Dionex Ultimate 3000 UHPLC system with a PDA detector controlled by Chromeleon 7.2 software coupled to a Q-Orbitrap high-resolution exactive focus mass spectrometer (Thermo Fisher Scientific, Bremen, Germany) was employed. The chromatographic system was coupled to MS with a source II heated electro-nebulization ionization probe (HESI II). The Q Exactive 2.0 SP 2, Xcalibur 2.3, and Trace Finder 3.2 software programs (Thermo Fisher Scientific) were used for UHPLC mass spectrometer control and data processing.

#### 4.4.1. Liquid Chromatography Parameters

As explained above, a portion of each extract (5 mg) obtained was dissolved in 5 mL of 1% formic acid–MeOH solution. It was then filtered through a 0.45 mm membrane (PTFE, Milford, MA, USA) and injected into the system. Liquid chromatography was performed using a UHPLC C-18 column (Acclaim, 150–length-4.6 mm ID, 5 m; Restek Corporation, Bellefonte, PA, USA) operated at 25 °C. The detection wavelengths were 255, 280, 355, and 640 nm, and the detection was performed from 200 to 800 nm. The mobile phases were 1% formic aqueous solution (A) and acetonitrile (B). The gradient program was as follows: (0.00 min, 5% B); (5.00 min, 5% B); (10.00 min, 30% B); (15.00 min, 30% B); (20.00 min, 70% B); (25.00 min, 70% B); (35.00 min, 5% B), and 12 min was allowed for balance before each injection. The flow rate was 1.0 mL/min and the injection volume was 10 µL. Standards and extracts dissolved in methanol were maintained at 10 °C during storage.

#### 4.4.2. Mass Spectrometry Parameters

Optimal operating conditions were achieved with the following parameters: gas pressure: 32 psi; auxiliary gas flow: 7 L min^−1^; flow rate of scanning gas: 1 L/min; spray voltage: −2500 V for negative ionization mode; capillary temperature: 320 °C; vaporizer temperature: 295 °C; and RF level of the S lens: 50%. High-resolution mass spectrometry (HRMS) was performed in full MS mode (resolution 70,000 FWHM at 200 Da) in the mass range *m*/*z* of 80–1000 for the negative ionization mode, and was performed to measure the target ions of precursors. The complete MS^2^ ions (full scan and data-dependent MS/MS mode) simultaneously recorded the MS/MS spectra (fragmentation) for the precursors. Maximum and minimum gain automatic control (AGC) objectives were 8 × 10^3^ and 5 × 10^3^, respectively, and the normalized collision energy was 30%. All parameters of the UHPLC-HRMS system were controlled through Thermo Scientific Xcalibur version 4.0 software (Thermo Scientific, Bremen, Germany).

To analyze the results of UHPLC-Q/Orbitrap/MS/MS, the Thermo Xcalibur 2.3 program was used to check the retention time, total mass, mass of fragments, molecular formula of possible compounds, and UV spectral data.

### 4.5. Anthelmintic Activity

#### 4.5.1. Larval Exsheathment Inhibition Test

Larval and egg tests were performed in accordance with the protocols reported by Reference [100]. For the LEI tests, eight treatments (40, 20, 10, 5, 2.5, 1.25, 0.6, and 0.3 mg/mL), six repetitions per treatment, and two controls (DMSO and Tween 80) were established. The three extracts were diluted in 1% DMSO and sterilized distilled water (pH: 6.9). Subsequently, the mixtures were prepared at concentrations of 40, 20, 10, 5, 2.5, 1.25, 0.6, and 0.3 mg/mL; next, at 3000 µL of diluted extract, Tween 80 (2.5%) was added to a solution containing 600 L_3_
*H. contortus* and sterilized distilled water (pH: 6.9) until a final volume of 6000 µL was reached. The control treatments involved DMSO (1%) and Tween 80 (2.5%). The mixtures were incubated in Falcon tubes at a temperature of 22 °C for 3 h in a Memmert IF55 incubator, with stirring every 30 min. To remove the extract, the tubes were centrifuged several times at 1950× *g* for 3 min in EBA 20 Hettich equipment, removing and replacing the supernatant.

Next, the exsheathment tests were prepared in 24 well flat-bottomed cell culture plates by adding solution containing 100 larvae along with sodium hypochlorite at a final concentration of 0.07% and sterilized distilled water (pH: 6.9) until a volume of 2000 µL was reached. Lugol was finally added to each column of the wells every 10 min, until a total of six additions had been made. The numbers of larvae with and without a sheath were counted under an Olympus CKX41 inverted microscope at 10×. The results are expressed in the form of percentage of exsheathment at each extract concentration.

#### 4.5.2. Egg Hatch Test

Stool was taken directly from the rectum of a sheep with monospecific infection of a Colombian isolate of *H. contortus* (ROCUB-2018), which is a field strain susceptible to benzimidazoles and resistant to levamisole. It was then macerated and filtered through four sieves (Fisherbrand™, Thermo Fisher Scientific Inc., Waltham, MA, USA) of different sizes. To facilitate filtration, sterilized distilled water (pH: 6.9) was used and the material retained in the smaller-pore-size sieve was recovered in 50 mL Falcon tubes. All of the material was then centrifuged at 459× *g* for 5 min, after which coprological syrup was added to the obtained sediment and centrifuged at 459× *g* for 5 min on Rotina 420 Hettich equipment. The supernatant was then deposited in the smallest sieve and sterilized distilled water (pH: 6.9) was added until the syrup was removed.

For the bioassay, 24 well plates were used, with eight treatments and six repetitions per treatment performed. The amount of extract needed to obtain concentrations of 40, 20, 10, 5, 2.5, 1.25, 0.6, and 0.3 mg/mL per well was diluted in 1% DMSO, and Tween 80 (2.5%) was added to the solution that contained 100 eggs and sterilized distilled sterilized water (pH: 6.9), for a total volume of 1000 µL. The control treatments contained DMSO (1%), Tween 80 (2.5%), and 3% fenbendazole (98%; Sigma^®^). The plates were incubated at 27 °C for 24 h in a Memmert IF55 incubator (Memmert GMbH, Buchenbach, Germany); Lugol was then applied to stop the process and the numbers of eggs, eggs with larvae, and larvae were determined using an inverted microscope.

To obtain the percentages of the tests described above, the formulas applied were used according to [51].

### 4.6. Statistical Analysis

For the data processing, mixed general linear models were used, which incorporated heteroscedastic variance models to assess the effect of extracts and fractions on LEI and EHI, and the repetitions were used as a random effect. When applying these models, the assumptions of normality and homogeneity of variance were evaluated by means of diagnostic charts of residuals with box-and-whisker plots, qq-plots, histograms, and dispersion of residuals compared with what was predicted. Likewise, significant differences were evaluated through the 5% Fisher LSD test. These analyses were processed using Infostat statistics software (Universidad Nacional de Cordoba-Cordoba, Argentina) [101], as well as through the platform for general and mixed linear models of the statistical program R statistics software (University of Auckland-Auckland, New Zaland) (version 3.4.4) [102]. To calculate the IC_50_ and IC_99_, the Probit model of the Statgraphics 2009 statistical package was used.

## 5. Conclusions

In order to evaluate the in vitro AA of the extracts of leaves of *G. sepium*, *L. leucocephala*, and *P. dulce*, LEIT and IHT were applied, and the compounds with potential AA were identified by analysis of the extracts using UHPLC-Q/Orbitrap/MS/MS. Through comparative evaluation of the ethanol extracts of the three most commonly used forage species in dry tropical forest, it was established that *G. sepium* showed a better effect on LEI of L_3,_ larval development, and EHI of samples from sheep infected with *H. contortus*. In addition, it showed the lowest IC_50_ values for LEI and EHI, proving its more potent anthelmintic effects. The results of UHPLC-Q/Orbitrap/MS/MS analysis supported the presence of major compounds exerting AA in the three plant species, among which were glycosylated flavonoids, flavonoids, phenylpropanoids other than flavonoids, methoxyphenols, anthraquinonic glycosides, amino acids, phenolic acids, glycosylates, and fatty acids. The greater diversity of compounds with anthelmintic activity in *G. sepium* could have influenced the results; however, it will be necessary to consider the interactions between compounds with AA in further studies. Compounds with AA were identified for the first time in all three studied plant species, although such compounds had previously been reported in other plant species. It is highly likely that these compounds exert synergistic effects within the same extract, and possibly among extracts from different plant species. The three species evaluated showed AA and thus represent a locally available resource alternative to conventional parasitic management in dry tropical conditions.

## Figures and Tables

**Figure 1 molecules-25-03002-f001:**
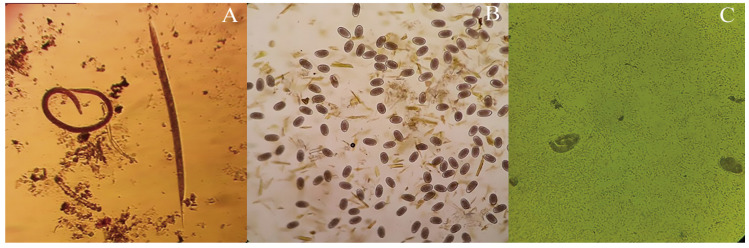
(**A**) *Haemonchus contortus* larvae with and without cuticle (40×); (**B**) morulae eggs of *Haemonchus contortus* (10×); (**C**) larvated eggs of *Haemonchus contortus* (40×).

**Figure 2 molecules-25-03002-f002:**
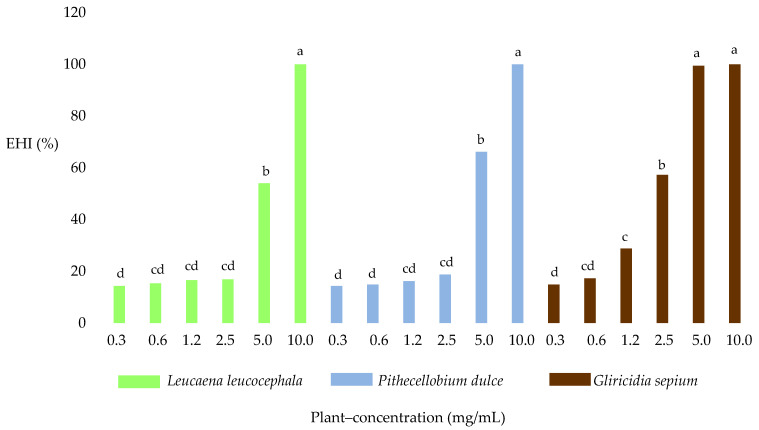
Effect of plant–concentration on the percentage of egg hatch inhibition (EHI). Means marked with the same letter are not significantly different (*p* > 0.05). Fisher LSD test.

**Figure 3 molecules-25-03002-f003:**
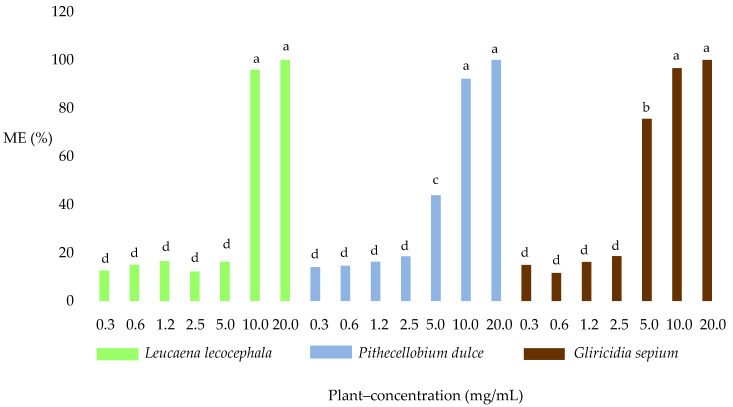
Plant–concentration effect on the percentage of morulated eggs (ME). Means marked with the same letter are not significantly different (*p* > 0.05). Fisher LSD test.

**Figure 4 molecules-25-03002-f004:**
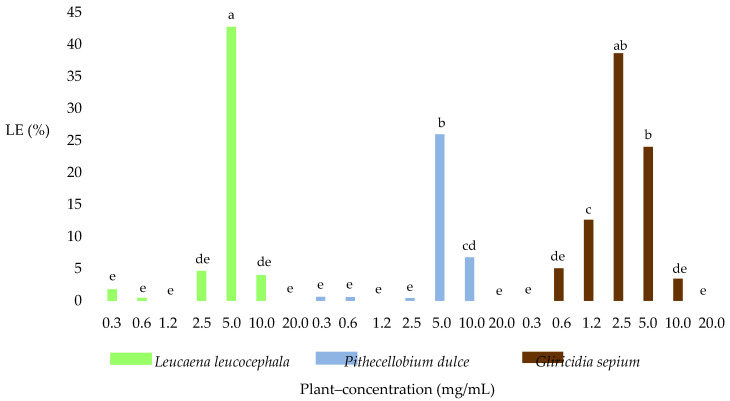
Effect of plant–concentration on the percentage of larvated eggs (LE). Means with a common letter are not significantly different (*p* > 0.05). Fisher LSD test.

**Table 1 molecules-25-03002-t001:** Secondary metabolites associated with anthelmintic activity in *Gliricidia sepium*, *Leucaena leucocephala*, and *Pithecellobium dulce*.

Classification	*Gliricidia sepium*	*Leucaena leucocephala*	*Pithecellobium dulce*
Glycosylated Flavonoids	Apigenin-di-C-dihexose-*O*-deoxyhexose *	Myricetin-3-*O*-hexoside *	Quercetin-3-glucoside * (Isoquercitrin)
	Apigenin-di-C-dihexose-*O*-deoxyhexose isomer *	Myricetin-3-arabinoside	Luteolin-7-*O*-glucoside * (Cynaroside or Glucoluteolin)
	Apigenin-7-*O*-Glucoside *	Myrcetin rhamnose derivative	Kaempferol-3-*O*-rhamnoside (Afzelin)
	Rutin*		
		Quercetin-3-*O*-arabinoside	
		Quercetin-3-*O*-pentoside *	
		Quercetin 3-*O*-rhamnoside	
		Kaempferol-3-*O*-pentoside *	
		Kaempferol-3-*O*-rhamnoside *	
Flavonoids		Luteolin *	
		Quercetin *	
		Apigenin *	
		Chrysoeriol *	
Phenylpropanoids other than Flavonoids	*p*-coumaroyl hexose *		
	Caffeoyl hexoside *		
	Dihydro-*p*-coumaric acid isomer *		
	*p*-coumaric acid		
	Leu/dihydro-p-coumaric acid *		
	Phe /Dihydro-p-coumaric acid *		
	p-Coumaric acid derivative *		
Methoxyphenols	Dihydrocapsiate *	Syringaldehyde syringate or derivative of quinic acid *	Dihydrocapsiate *
		Syringaldehyde syringate or derivative of quinic acid *	
Anthraquinonic Glycosides	Trihydroxyanthraquinone-*O*-methylgluconate-glucoside *		
	Trihydroxyanthraquinone-*O*-methylgluconate-deoxymethylgluconic *		
	Trihydroxyanthraquinone-*O*-methylgluconate *		
	Trihydroxyanthraquinone-*O*-methylgluconate isomer *		
Amino acids	Glutamic acid *		
	*N*-Carbobenzyloxy-*l*-isoleucine *		
Glycosylated phenolic acids	Phenyllactic acid-2-*O*-glucoside *		
Fatty acids	Ázelaic acid *		Azelaic acid *

* Compounds identified for the first time in each plant.

**Table 2 molecules-25-03002-t002:** Comparison of the effect of the three forage species on larval exsheathment inhibition in vitro.

Plant Extract	Average and Standard Error of Larval Exsheathment Inhibition (%)				
	40 mg/mL	20 mg/mL	10 mg/mL	Control Tween	Control DMSO
*Gliricidia sepium*	91.28 ± 2.2 ^a^	45.14 ± 2.0 ^b^	8.89 ± 1.9 ^c^	2.08 ± 2.0 ^de^	5.01 ± 1.8 ^cde^
*Leucaena leucocephala*	41.01 ± 18.9 ^bc^	21.42 ± 5.6 ^c^	9.62 ± 1.7 ^c^	2.06 ± 1.0 ^de^	4.89 ± 1.1 ^cde^
*Pithecellobium dulce*	44.66 ± 20.5 ^bc^	23.35 ± 5.9 ^c^	7.44 ± 1.0 ^c^	2.04 ± 0.68 ^e^	4.82 ± 0.7 ^cde^

Values with the same letters not significantly different (*p* > 0.05).

**Table 3 molecules-25-03002-t003:** IC_50_ and IC_99_ of extracts for larval exsheathment inhibition (LEI) and egg hatch inhibition (EHI).

Plant Extracts	Test	IC_50_ (mg/mL)	Lower Limit Confidence Level 95.0%	Upper Limit Confidence Level 95.0%	IC_99_ (mg/mL)	Lower Limit Confidence Level 95.0%	Upper Limit Confidence Level 95.0%
*Gliricidia sepium*	*LEI*	22.44	20.50	24.74	65.44	59.18	73.47
	*EHI*	1.97	1.76	2.20	5.46	4.89	6.23
*Leucaena leucocephala*	*LEI*	43.35	37.99	50.98	101.05	86.63	122.52
	*EHI*	4.31	3.90	4.79	11.45	10.29	12.98
*Pithecellobium dulce*	*LEI*	41.77	36.39	49.48	103.03	87.83	125.88
	*EHI*	3.91	3.53	4.35	10.36	9.29	11.79

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
