# Peer review of "In Vitro Anthelmintic Evaluation of Gliricidia sepium, Leucaena leucocephala, and Pithecellobium dulce: Fingerprint Analysis of Extracts by UHPLC-Orbitrap Mass Spectrometry"

_molecules, 2020, doi:10.3390/molecules25133002_

Round 1
Reviewer 1 Report
Dear Editor and authors,
The work represents an original contribution that describes the activity of alcoholic extracts from three species of plants that could be used as a vermifuge in sheep farming. In addition to the toxicological tests, the authors identified the secondary compounds present in the extracts and discussed their biological functions.
The results presented are consistent and robust. However, data presentation needs to be improved before recommending the acceptance of the manuscript.
Below are the suggestions for improving the article.
Keywords: Species names in italic;
Introduction:
Lines 41-43 - The problem should be addressed globally. I believe that this is the case for gastrointestinal parasites;
L45 – Change Trichostrongilidae to Trichostrongylidae;
The authors are talking about nematode parasitism. The parasitism type should be made evident;
Objectives - one of the authors' objectives was: "ii) to identify the bioactive compounds in all active extracts using UHPLC-77 Q / Orbitrap / MS / MS. "
There was no separate analysis of the activity of each identified compound. Why were these compounds classified a priori as bioactive?
Material and Methods:
L92 – Please, provide the number of the Whatman filter paper;
No information on the purity of the water used in aqueous solutions is provided in the methodology. As it is an essential source of organic contamination, please provide these details.
L132 - Please specify the eight treatments. Were the different concentrations?
Results:
Table 1 – Considering all compounds are secondary metabolites, include this information in the tile and eliminate it from the first line of the table. Let only the name of each plant species.
Fig. 1 A) – Larvae with cuticles are identified with a black arrow. There is no problem in identifying the larvae without cuticle with a different arrow to be more didactic;
Table 2 – in a final version, this table should not be broken;
Figure 2 – The green block is covering the plant name "Leucaena leucocephala";
Figure 2 and 3 – Please, specify the statical used to multiple comparisons in the legend os each figure;
Figure 4 – there is a problem with the legend of this figure; the name of the second plant species is missing;
L285 - As a matter of style, it is not advised to authors to start a sentence with an abbreviation of species. "G. sepium"
Author Response
- Keywords:Species names in italic;
Answer: We appreciate the suggestion and changes were made in the manuscript.
- Lines 41-43 - The problem should be addressed globally. I believe that this is the case for gastrointestinal parasites;
Answer: We appreciate the suggestion and changes were made in the manuscript.
- L45 – Change Trichostrongilidae to Trichostrongylidae;
Answer: We appreciate the suggestion and changes were made in the manuscript.
- The authors are talking about nematode parasitism. The parasitism type should be made evident;
Answer: We appreciate the suggestion and changes were made in the manuscript.
- Objectives- one of the authors' objectives was: "ii) to identify the bioactive compounds in all active extracts using UHPLC-77 Q / Orbitrap / MS / MS. "
There was no separate analysis of the activity of each identified compound. Why were these compounds classified a priori as bioactive?
Answer: In fact, the analysis by UHPLC-Q-Orbitrap / MS / MS led to the identification of compounds in the three extracts and subsequently their anthelmintic activity was investigated and compared with what was reported in the literature, finding for each case those listed in Table 1. observation made is very pertinent and the respective adjustment to the objective was made.
Material and Methods:
- L92 – Please, provide the number of the Whatman filter paper;
No information on the purity of the water used in aqueous solutions is provided in the methodology. As it is an essential source of organic contamination, please provide these details.
Answer: We appreciate the suggestion and changes were made in the manuscript.
- L132 - Please specify the eight treatments. Were the different concentrations?
We appreciate the suggestion and changes were made in the manuscript.
Results:
- Table 1 – Considering all compounds are secondary metabolites, include this information in the tile and eliminate it from the first line of the table. Let only the name of each plant species.
Answer: We appreciate the suggestion and changes were made in the manuscript.
- 1 A) – Larvae with cuticles are identified with a black arrow. There is no problem in identifying the larvae without cuticle with a different arrow to be more didactic;
Answer: We appreciate the suggestion and changes were made in the manuscript.
- Table 2 – in a final version, this table should not be broken;
Answer: We appreciate the suggestion and changes were made in the manuscript.
- Figure 2 – The green block is covering the plant name "Leucaena leucocephala";
Answer: We appreciate the suggestion and changes were made in the manuscript.
- Figure 2 and 3 – Please, specify the statical used to multiple comparisons in the legend os each figure;
Answer: We appreciate the suggestion and changes were made in the manuscript.
- Figure 4 – there is a problem with the legend of this figure; the name of the second plant species is missing;
Answer: We appreciate the suggestion and changes were made in the manuscript.
- L285 - As a matter of style, it is not advised to authors to start a sentence with an abbreviation of species. "G. sepium"
Answer: We appreciate the suggestion and changes were made in the manuscript.

Reviewer 2 Report
In this study, the anthelmintic activity of ethanolic extracts obtained from Gliricidia sepium, Leucaena leucocephala, and Pithecellobium dulce was evaluated using the third-stage larval (L3) exsheathment inhibition test and egg hatch test on Haemonchus contortus. The study seems to give some useful data on the topic. The study can be considered for the Journal after revisions. Below I provide some points for the revision.
Comments:
Title: In vitro- should be in italic. Latin name should also be italic: Gliricidia sepium, Leucaena leucocephala, Pithecellobium dulce
Second part of the title is with smaller letters. Not easy to understand ‘UHPLC-4
Q/Orbitrap/MS/MS’ change it for a more Reader friendly form.
Abstract
L30-34: Long sentence. Divide it.
Keywords:
L37: Again, latin name should be italic: Gliricidia sepium, Leucaena leucocephala, Pithecellobium dulce, Haemonchus contortus
M and M
L84:Why you use full name here if you already introduced them before: G. sepium, L. leucocephala, and P. dulce?
Results
L230: Give in full EHI. Green colour for Leucaena is not in the right place.
L244: Give in the title ‘(ME)’
L264: Give in the title ‘(LE)’. Check blue colour note ..
L272: Give for the title LEI and EHI. … for Larval exsheathment inhibition (LEI) and eggs hatch inhibition (EHI).
Discussion
L338: Delete (Santos et al, 2017)
L395: Give Consclusions in points.
L398: References. Check journal names: short or full form? e.g. L400,L412, L420, L632
Author Response
- Title: In vitro- should be in italic. Latin name should also be italic: Gliricidia sepium, Leucaena leucocephala, Pithecellobium dulce
Answer: We appreciate the suggestion and changes were made in the manuscript.
- Second part of the title is with smaller letters. Not easy to understand ‘UHPLC-Q/Orbitrap/MS/MS’ change it for a more Reader friendly form.
Answer: We appreciate the suggestion and changes were made in the manuscript.
- L30-34: Long sentence. Divide it.
Answer: We appreciate the suggestion and changes were made in the manuscript.
Keywords:
- L37: Again, latin name should be italic: Gliricidia sepium, Leucaena leucocephala, Pithecellobium dulce, Haemonchus contortus
Answer: We appreciate the suggestion and changes were made in the manuscript.
- L84:Why you use full name here if you already introduced them before: G. sepium, L. leucocephala, and P. dulce?
Answer: We appreciate the suggestion and changes were made in the manuscript.
Results
- L230: Give in full EHI. Green colour for Leucaena is not in the right place.
We appreciate the suggestion and changes were made in the manuscript.
- L244: Give in the title ‘(ME)’
We appreciate the suggestion and changes were made in the manuscript.
- L264: Give in the title ‘(LE)’. Check blue colour note.
Answer: We appreciate the suggestion and changes were made in the manuscript.
- L272: Give for the title LEI and EHI. … for Larval exsheathment inhibition (LEI) and eggs hatch inhibition (EHI).
Answer: We appreciate the suggestion and changes were made in the manuscript.
Discussion
- L338: Delete (Santos et al, 2017)
Answer: We appreciate the suggestion and changes were made in the manuscript.
- L395: Give Consclusions in points.
Answer: We appreciate the reviewer's suggestion, however, we feel that it is not appropriate.
- L398: References. Check journal names: short or full form? e.g. L400,L412, L420, L632
Answer: We appreciate the suggestion and changes were made in the manuscript. Other references were corrected.
